# Structural and Spectroscopic Study of New Copper(II) and Zinc(II) Complexes of Coumarin Oxyacetate Ligands and Determination of Their Antimicrobial Activity

**DOI:** 10.3390/molecules28114560

**Published:** 2023-06-05

**Authors:** Muhammad Mujahid, Natasha Trendafilova, Georgina Rosair, Kevin Kavanagh, Maureen Walsh, Bernadette S. Creaven, Ivelina Georgieva

**Affiliations:** 1Centre of Applied Science for Health, TU Dublin, Tallaght, D24 FKT9 Dublin, Ireland; 2Institute of General and Inorganic Chemistry, Bulgarian Academy of Sciences, 11 Acad. G. Bonchev Str., 1113 Sofia, Bulgaria; ntrend@svr.igic.bas.bg; 3School of Engineering & Physical Sciences, Heriot-Watt University, Edinburgh EH14 4AS, UK; 4Department of Biology, Maynooth University, W23 F2K8 Maynooth, Ireland; kevin.kavanagh@mu.ie; 5School of Chemical and Pharmaceutical Sciences, TU Dublin, Central Quad, Grangegorman, D07 H6K8 Dublin, Ireland

**Keywords:** coumarin, Zn(II) and Cu(II) complexes, DFT modelling, antimicrobial activity, X-ray structure

## Abstract

Tackling antimicrobial resistance is of increasing concern in a post-pandemic world where overuse of antibiotics has increased the threat of another pandemic caused by antimicrobial-resistant pathogens. Derivatives of coumarins, a naturally occurring bioactive compound, and its metal complexes have proven therapeutic potential as antimicrobial agents and in this study a series of copper(II) and zinc(II) complexes of coumarin oxyacetate ligands were synthesised and characterised by spectroscopic techniques (IR, ^1^H, ^13^C NMR, UV-Vis) and by X-ray crystallography for two of the zinc complexes. The experimental spectroscopic data were then interpreted on the basis of molecular structure modelling and subsequent spectra simulation using the density functional theory method to identify the coordination mode in solution for the metal ions in the complexes. Interestingly, the solid-state coordination environment of the zinc complexes is in good agreement with the simulated solution state, which has not been the case in our previous studies of these ligands when coordinated to silver(I). Previous studies had indicated excellent antimicrobial activity for Ag(I) analogues of these ligands and related copper and zinc complexes of coumarin-derived ligands, but in this study none of the complexes displayed antimicrobial activity against the clinically relevant methicillin-resistant *Staphylococcus aureus* (MRSA), *Pseudomonas aeruginosa* and *Candida albicans*.

## 1. Introduction

Antimicrobial resistance (AMR) is of increasing concern worldwide and has been consistently highlighted by the World Health Organisation as one of the biggest global health challenges facing us in the twenty-first century [1]. In the European Union, AMR causes 25,000 deaths and 2.5 million extra hospital days per year [2]. Whilst the overuse of antibiotic and antifungal agents has long been identified as one of the main causes of the rise in antimicrobial resistance, what is of increasing concern is the emergence of new resistance mechanisms by pathogenic bacteria and fungi. The identification of new resistance mechanisms may be due to increased research into AMR, an area prioritised by many governments and international agencies, or may be as a result of the increasing use of last-line of defence antibiotics and antifungals [2]. Essentially, a vicious cycle may be emerging, necessitating the development of new antimicrobial agents with alternative targets to existing therapies. *Pseudomonas aeruginosa* is included in the World Health Organization (WHO) list of critical priority pathogens and is an emerging threat to global health owing to the rapid rise of convergent AMR and hypervirulent strains. It poses an emerging threat to global health, by causing high morbidity and mortality rates, making treatments difficult [3,4]. *Methicillin resistant Staphylococcus aureus,* MRSA, is on the same list but ranked as a high priority pathogen, and *Candida albicans* is on the WHO list of priority fungal pathogens. All three pathogens have previously shown susceptibility to both zinc and copper complexes and this susceptibility has led to this present study [5,6].

Our group has been involved in the isolation of coumarin-based therapeutics for a number of years and have reported a series of such compounds with excellent antimicrobial and antifungal activity [7,8,9,10,11,12]. Coumarins are a class of compounds which have benzopyrone as their central unit. They have proven applications as antibiotics, anticoagulants, fungicides and anti-inflammatories. Recent reviews have highlighted how versatile a scaffold coumarin is for bioactive compounds and details of the many applications of the derivatised coumarins are contained therein [13,14,15]. The early antimicrobial studies on substituted coumarins focused on 4-hydroxy- and 4-oxy-substituted derivatives [16,17,18,19,20] as these coumarin derivatives showed particularly high antimicrobial activity. A recent review has also summarized the antimicrobial activity of 4-substituted coumarins in general [21]. Our previous studies have shown that Ag(I) complexes of coumarin-3-carboxylates and coumarin-4-carboxylates have excellent activity against *MRSA* and *P. aeruginosa*, respectively [8,22], but solubility in aqueous solution was poor. A later series of Ag(I) complexes isolated with oxyacetate coumarin derivatives showed improved solubility and had good activity against a range of microbial species [7]. We were interested in extending the position of the oxyacetate moiety on the coumarin nucleus and also the nature of the substituent on the aromatic ring. In previous studies, variation in the substituent on the aromatic ring did cause significant variation in activity between the bacterial and fungal pathogens, with only Ag(I) complexes of ligands **4** and **8** of this study showing antifungal activity whilst all other Ag(I) complexes were active against both bacterial strains tested.

The need for a strategic approach towards tackling antimicrobial resistance has resulted in several other metal-based compounds being investigated as novel therapeutics including complexes of zinc and copper ions. Four homoleptic zinc and copper complexes of Schiff base ligands derived from 2-(3,4-dimethoxyphenyl)ethan-1-amine showed excellent antimicrobial activity against pathogenic antibacterial strains *Escherichia coli, S. aureus* and the antifungal strain, *C. albicans* [23]. A number of other zinc complexes have also shown excellent antimicrobial activity, including Zn(II) complexes of chlorogenic acid *p*-coumaric acid, and ferulic acid [24], which demonstrated antibacterial activity against *S. aureus*, while Zn(II) complexes with aromatic nitrogen-containing heterocycles, [ZnCl_2_(qz)_2_], [ZnCl_2_(1,5-naph)]*_n_* and [ZnCl_2_(4,7-phen)_2_], (qz is quinazoline, 1,5-naph is 1,5-naphthyridine and 4,7-phen is 4,7-phenanthroline) have shown good antifungal activity against two *Candida* strains (*C. albicans* and *Candida parapsilosis*) [25]. A zinc carboxylate complex has shown fungicidal and bactericidal activity [26] and a zinc complex of a coumarin-glycine conjugate showed higher antifungal activity than a commercial drug Greseofulvin and had reasonable antimicrobial activity against *Escherichia coli* and *Bacillus cirroflagellosus* [27]. More recently, Cu(II) and Zn(II) complexes of the clinically used antifungal drug fluconazole, {[CuCl_2_(fcz)_2_]^.^5H_2_O}_n_, and {[ZnCl_2_(fcz)_2_]·2C_2_H_5_OH}_n_, have been tested and demonstrated excellent antimicrobial activity against *C. albicans and P. aeruginosa* [28]. The solubility of the Cu(II) and Zn(II) complexes of coumarin-3-carboxylate that we previously studied was poor and their low antimicrobial properties were thought to be related to their limited solubility in anything other than DMSO [29]. However, we have previously reported the anti-*Candida* activity of a series of coumarin-derived and quinolone-2(1*H*)-one triazole derived Schiff base Cu(II) complexes whereby the complexes exhibited anti-*Candida* activity comparable to that of the commercial antifungal drugs ketoconazole and amphotericin B [30]. More recently, a series of coumarin-thiadiazole hybrids and their Cu(II) and Zn(II) complexes showed potent antimicrobial activity [31].

This paper reports on the synthesis of Zn(II) and Cu(II) complexes with a series of 10 coumarin oxyacetate ligands, the Ag(I) complexes of which we previously reported [7]. The new complexes were characterised by X-ray crystallographic studies (for two Zn(II) complexes), IR, NMR and UV-Vis spectroscopies as well as by DFT/TDDFT calculations. The antimicrobial activity of the Zn(II) and Cu(II) complexes was assessed and compared to the proven activity of their Ag(I) analogues [7] and the reported activity of similar Zn(II) and Cu(II) complexes. Molecular geometries of the new Zn(II) and Cu(II) complexes were simulated in solid state and in solution by DFT model calculations. The comparison has shown that the studied M(II) complex coordination geometries in solution are the same as those found experimentally in the solid state (six-coordinated metal(II) centre in pseudo-octahedral geometry). This finding is in contrast to the Ag(I) analogues, which have shown different molecular geometries in solution (4-coordinated metal(I) centre and close to planar geometry) and in the solid state (extensive contacts to Ag(I) and five-coordinated metal(I) centre). The experimental and DFT-simulated spectra of the new Zn(II) and Cu(II) complexes (IR, UV-Vis, NMR) were correlated to explain their spectroscopic behaviour and their likely form in biological matrices.

## 2. Results and Discussion

### 2.1. Synthesis

#### 2.1.1. Synthesis of Ligands

The synthesis and characterisation of all the ligands (**1**–**10**, Appendix A) used in this study was previously published by our group [7].

#### 2.1.2. Synthesis of Zn(II) (**11**–**20**) and Cu(II) (**21**–**30**) Complexes of 2-(2-oxo-2H-chromen-substituted-yl)oxy Acetate

The 2-(2-oxo-2*H*-chromene-substituted-yl)oxy acetic acids (**1–10**) (1 mmol) were dissolved in hot absolute ethanol (20 mL). Metal(II) acetate salts (anhydrous zinc acetate and copper acetate mono-hydrate (0.5 mmol)) were dissolved in deionised water (5 mL) at room temperature and mixed immediately with the appropriate ligand dissolved in hot ethanol. The resulting mixture was heated at reflux temperature for 3 h, then it was allowed to cool. The resulting precipitate was filtered off, washed with deionised water (20 mL × 2) and absolute ethanol (10 mL × 2) and finally dried in a vacuum oven for 3 days at 80 °C. All Zn(II) complexes (**11**–**20**) were white in colour, whereas the Cu(II) complexes (**21**–**30**) were differently coloured. The analytical data for Zn(II) and Cu(II) complexes are given in Table 1. Structural, IR, UV-Vis and NMR data are given in Table 2, Table 3, Table 4, Table 5 and Table 6.

### 2.2. Characterisation of Zn(II) (**11**–**20**) and Cu(II) (**21**–**30**) Complexes

A series of Zn(II) complexes (**11**–**20**) and Cu(II) complexes (**21**–**30**) were synthesised by ligand–anion exchange reaction between a simple metal(II) salt and appropriate 2-(2-oxo-2*H*-chromen-substituted-yl)oxy acetic acid ligands (**1**–**10**) (Appendix A). The reaction methodology is presented in Figure 1.

The complexes were characterized by decomposition temperatures, microanalysis, atomic absorption spectroscopy, thermogravimetric analysis, conductivity and magnetic moment measurements, IR, UV-Vis and ^1^H, ^13^C NMR spectroscopies and X-ray crystallography. The complexes obtained were prepared in good yields ranging from 60% to 72%. They showed good solubility in DMSO at ambient temperature and sparing solubility in ethanol:water 50:50 mixture at 60 °C. Analytical data for the complexes given in Table 1 were within an acceptable range of the theoretical values. The metal content for these complexes (Table 1) were estimated via atomic absorption spectroscopy and were later confirmed by thermogravimetric analysis (TGA). The obtained results showed a 1:2 ratio between metal-to-ligand along with four water molecules in each of the Zn(II) and Cu(II) complexes. The thermograms recorded for the Zn(II) and Cu(II) complexes (in this study) revealed identical thermal decomposition behaviour of the 2-[2-oxo-2*H*-chromen-3-yl]oxy acetic acid-based Zn(II) and Cu(II) complexes, Appendix A. Thermogravimetric analysis of the metal(II) complexes studied showed the loss of four water molecules in a single step transition at the temperature range ca. 80 to 250 °C. The significant mass loss in a single step and at relative higher temperature range indicated that possibly all of the water molecules were of the same type, i.e., water of coordination. However, Karaliota et al. [32] have suggested the loss of both type water molecules of coordination and crystallisation in a single step transition. In their complex, lattice water molecules created an extensive hydrogen-bonded network which could account well for their higher removal temperature, probably overlapping with the removal temperature of coordinated water molecules. All of the Zn(II) and Cu(II) complexes showed a significant loss in mass after 270–285 °C, representing decomposition of the coumarin-based ligand. Finally, the complexes decomposed gradually with the formation of metal-oxide above 450–500 °C. The TGA data obtained along with microanalytical and atomic absorption data were used to deduce the empirical formulae for these complexes (Table 1).

The molar conductivity values of Zn(II) and Cu(II) complexes in DMSO at 25 °C were in the range 1.11 to 2.97 Scm^2^mol^−1^ and at 37 °C (physiological temperature) these values were slightly higher: 1.33 ÷ 3.08 Scm^2^mol^−1^ (Appendix A). The conductivity values at both temperatures were too low to account for any dissociation of the metal complexes in DMSO, hence these complexes were considered nonelectrolyte in nature, similar to other reported coumarin- and metal-based complexes [7,8,9,10,11].

### 2.3. Crystal structure of [Zn(C-4oxy-acet)_2_(H_2_O)_4_] (**12**) and [Zn(4CF_3_-C-7oxy-acet)_2_(H_2_O)_4_] (**19**)

Among the series studied, for two of the coordination compounds, [Zn(C-4oxy-acet)_2_(H_2_O)_4_] (**12**) and [Zn(4CF_3_-C-7oxy-acet)_2_(H_2_O)_4_] (**19**), diffraction quality single crystals were obtained by recrystallisation from ethanol–water (50:50) solution mixture. Despite repeat attempts to isolate crystals of the copper complexes 21–30, crystals of suitable quality for XRD could not be obtained. It is notable that no other coumarin metal complexes were found in the March 2022 version of the Cambridge Structural Database [33], which suggests that they are challenging to crystallize.

Crystal data and structure refinement details of complexes **12** and **19** are given in the Appendix A, respectively, and selected bond lengths and bond angles are presented in Table 2 and hydrogen bond parameters in Appendix A. The X-ray crystal structures of **12** and **19** together with the atom numbering scheme and the crystal packing structure diagrams are shown in Figure 1 and Figure 2, respectively. The crystal structures of the complexes revealed that they have a pseudo-octahedral coordination geometry, with Zn(II) ion having six contacts within its coordination sphere. In both **12** and **19,** Zn lies on inversion centres. Two of the contacts around Zn(II) ion were constituted by two deprotonated coumarin-based ligands C-4oxy-acet (**2**) for complex **12** and 4CF_3_-C-7oxy-acet (**9**) for complex **19** bonded in a monodentate manner and the remaining four contacts were with four water molecules. In the complexes **12** and **19**, the ligands **2** [Zn1-O3 distance 2.0732(9) Å, C1-O3-Zn1 angles 125.61(8)°] and **9** [Zn1-O2 distance 2.0486(15) Å, C4-O2-Zn angles 127.95(13)°], respectively, were coordinated to the Zn(II) ion via the deprotonated carboxylate group arranged in a *trans* orientation. The Zn(II)-O_carboxylate_ bond distances for these complexes were found to be slightly longer than the Zn(II)-O_carboxylate_ (2.0058 Å) for the reported crystal structure of aceto*bis*(L-arginine)Zn(II) acetate trihydrate by Nolan et al. [34]. However, Chen et al. reported comparable Zn(II)-O_carboxylate_ bond distances (2.082–2.095 Å) for one of their octahedral Zn(II) complexes derived from zwitterionic carboxylates [35].

The 1:2, metal-to-ligand ratio and the presence of four coordinated water molecules in complexes **12** and **19** were initially observed by atomic absorption spectroscopy, elemental and thermogravimetric analyses prior to recrystallisation. However, complex **19** showed two additional water molecules of crystallisation which were not observed in the powdered samples (Figure 2). The four water molecules were coordinated to the metal ion via the oxygen atoms. The Zn(II)-O_water_ bond distances (2.082–2.115 Å) were comparable to those reported for other octahedral Zn(II) complexes [35]. The axial water molecules in complex **12** [O1-H11*W*···O4, distance 1.85(2) Å (H---*A*)], O4-C1-O3 angles 125.65(12)°] and **19** [O1-H1B···O3, distance 1.847(10) Å [(H---*A*)], O3-C4-O2 angles 126.26(19)°] formed hydrogen bonding with the free oxygen atom of carboxylate groups and have longer Zn(II)-O_water_ bond distances than the equatorial water molecules. The array of coordinated atoms around the metal centre (Zn(II)) formed a pseudo-octahedral coordination geometry in both the complexes. Moreover, Zn-O-C-C torsional angles of 158.67(8), and 166.22(15), −166.17° were also observed in complexes **12** and **19**, respectively. These angles originated between two different planes constituted by Zn(II)-O_carboxylate_ and C_carboxylate_-C_methylene_ (Figure 1b and Figure 2b). Due to this torsional angle, the derivatised coumarin molecules are aligned along a diagonal [−1,0,1]. Coumarin molecules were observed to be planar in both of the complexes and the heterocycle C5O plane centroid to C6 phenyl plane centroid distance between intermolecular aromatic rings of complexes **12** and **19** was 3.7960(7) Å and 3.6597(10) Å, respectively. Two of the four water ligands on **19** are disordered (O2a 49%, O2b 51% relative occupancy).

### 2.4. Experimental and Theoretical IR Spectral Analysis of Zn(II) and Cu(II) Complexes of 2-(2-oxo-2H-chromen-Substituted-yl)oxy Acetic Acid Ligands. Magnetic Moment Data of Cu(II) Complexes

Selected experimental IR frequencies of the Zn(II) (**11**, **12** and **19**) and the Cu(II) (**22**) complexes studied, and the corresponding ligands **1**, **2** and **9**, are given in Table 3. In order to analyse the observed IR spectral data, calculations of the IR spectra and the relative weights of internal coordinates to the normal modes were performed for the [Zn(C-3oxy-acet)_2_(H_2_O)_4_] (**11)** and [Cu(C-4oxy-acet)_2_(H_2_O)_4_] (**22**) complexes and the relative **1**, **2** and **9** ligands at the molecular level. Interpretation of the observed IR spectral data of the complexes was not a trivial task since in the solid state (Figure 1 and Figure 2), strong intra- and intermolecular hydrogen bonds are formed (due to the presence of the electron-donor carboxylic and carbonylic groups and electron acceptor water hydrogens) and they specifically perturb the bond lengths of the compounds studied. The available X-ray structures of [Zn(C-4oxy-acet)_2_(H_2_O)_4_] **(12)** and [Zn(4CF_3_-C-7oxy-acet)_2_(H_2_O)_4_.2H_2_O] **(19)** give a possibility for more reliable calculations of the vibrational frequencies and IR intensities based on the model crystallographic structures **12** and **19** (Table 3). An inspection of calculated structural data for **12** in gas phase and in solid state showed that the solid state-calculated bond lengths are in better agreement with the experimental ones (Appendix A. The experimental IR spectra (2000–400 cm^−1^) of C-4oxy-acetH (**2**) and [Zn(C-4oxy-acet)_2_(H_2_O)_4_] **(12)** compared to the calculated one of the Zn(II) complex are shown in Appendix A. Similar IR spectra were expected for the crystalline and powder Zn(II) complexes and this supposition was confirmed by the IR spectra of both samples of [Zn(4CF_3_-C-7oxy-acet)_2_(H_2_O)_4_] **(19)** complex (Appendix A). The observed similar behaviour in the IR spectra of the Cu(II) (**21**–**30**) and the Zn(II) (**11–20**) complexes indicated that the coordination mode in the Cu(II) complexes is identical to that already found for the Zn(II) complexes of the same series of ligands.

The IR spectra of the free ligands exhibited a medium intensity band at ca. 3434 to 3420 cm^−1^, which is attributed to the ν(OH)_carboxylate_ vibration. In the spectra of the metal complexes, this band should disappear since the active form of the coordinating ligand is the deprotonated one. However, broad and medium intensity bands at ca. 3500 to 3400 cm^−1^ were observed (Table 3) and they were assigned to the ν(O-H) stretching vibrations of water molecules in the inner and outer coordination sphere of the complexes as it was indicated by the X-ray structures of **12** and **19** and by the thermogravimetric analysis. The coordination of water molecules was manifested also by the presence of a new band at ~ 816 cm^−1^ in the IR spectrum of the Zn(II) complex **12**, corresponding to the rocking (OH_2_) vibration, Appendix A. The two type stretching C=O vibrations (ν(C=O)_coum_ and ν(C=O)_COOH_)) of the ligands (**1–10**) appear as a doublet or a broad band with a very high intensity in the range 1713–1757 cm^−1^ and their order could be changeable depending on the involved hydrogen-bonded interactions.

According to the calculations, the lactone carbonyl stretching vibrations produce very intensive bands in the IR region 1750–1670 cm^−1^ for the Zn(II) and Cu(II) complexes studied. There is no clear trend of ν(C=O) frequency shift going from ligands to their Zn(II) and Cu(II) complexes because the ν(C=O) vibrational mode is perturbed in addition to various weak interactions (Table 3). For example, the vibrational band corresponding to the lactone carbonyl ν(C=O) mode of **12** appears at 1675 cm^−1^ and it is shifted to lower wavenumbers by ~ 50 cm^−1^ as compared to that of its neutral ligand (at 1720 cm^−1^). The wavenumber shift could be explained with the elongation of C=O bond (ligand **2** 1.214 Å (calc)→ complex **12** 1.230 (exp), 1.245 (calc)), which is produced by the intermolecular bonding (C=O_coum_···H_w_ ~ 1.93 Å (exp)) as is seen by X-ray structure of **12** (Figure 1). The IR spectrum of complex **19** showed the ν(C=O) band at 1727 cm^−1^. The higher ν(C=O) frequency for **19** (1727 cm^−1^) as compared to that of **12** (1675 cm^−1^) is in agreement with the C=O bond length 1.212 Å and 1.230 Å, respectively, and this result could be explained with weaker intermolecular (C=O···H ~2.40 Å) interactions in **19** than (C=O···H ~1.94 Å) in **12**.

The prediction and interpretation of the stretching vibrations of the carboxylic group, ν(COO)_as_ and ν(COO)_s_, also appeared complicated in the metal complexes studied due to the solid-state effect and intramolecular bonding of the uncoordinated carboxylic oxygen with the aqueous hydrogens. Moreover, these bands are medium, partially overlapped and not well indicative. The solid-state calculations of Zn(II) complexes **12** (**19)**, predicted that the ν(COO)_as_ and ν(COO)_s_ vibrations appear as strong band at 1603 (1567) cm^−1^ and medium band at 1340 (1347) cm^−1^ (Table 3). The predicted ν(COO) for the Cu(II) and Zn(II) complexes of ~ 220–300 cm^−1^ are consistent with the experimental data for other unidentate carboxylate metal complexes [36] and confirm the monodentate ligand binding to the Zn(II) and Cu(II) through the oxygen atom of the deprotonated carboxylate group.

The shorter Zn/Cu–O_COO_ than Zn/Cu–O_water_ bond lengths from the experiment (Table 2) generate ν(Zn/Cu–O) frequencies at ~385–413 cm^- 1^ versus to ~248–311 cm^−1^, according to the calculations (Table 3).

Further evidence of the mononuclear structure of the Cu(II) complexes is the room temperature magnetic moments of the powdered samples (Appendix A). The magnetic moment values (μ_eff_) of the Cu(II) complexes were in the range 1.72 to 1.92 B.M. which is consistent with the expected range of mononuclear Cu(II) complexes (μ_eff(theor)_= 1.73 B.M.) corresponding to the doublet state without Cu(II)-Cu(II) interactions and insignificant spin-orbit interactions. The magnetic values observed fall within the range generally observed for octahedral Cu(II) complexes [37] as well as for mononuclear Cu(II) complexes (1.75 to 2.20 B.M.).

### 2.5. Experimental and Theoretical UV-Vis Spectral Analysis of Zn(II) and Cu(II) Complexes of 2-(2-oxo-2H-chromen-Substituted-yl)oxy Acetic Acid Ligands

To establish the formation of the complexes and to obtain information about their geometric and electronic structures in solution, molecular structure modelling and spectra simulations were performed. The UV-Vis spectra of the 2-(2-oxo-2*H*-chromen-substituted-yl)oxy acetic acid-based ligands and their Zn(II) and Cu(II) complexes were compared and interpreted. The absorption spectral data, i.e., the maximum absorption wavelength (λ_max_ in nm) and molar absorptivity values (M^−1^ cm^−1^) for all of the complexes along with their ligands, are given in Table 4. The UV-Vis spectra in the range 260–400 nm of Zn(II) (**11–20**) and Cu(II) (**21–30**) complexes in DMSO are shown in Appendix A, respectively.

DFT/TDDFT molecular modelling calculations were performed for [Zn(C-4oxy-acet)_2_(H_2_O)_4_] **(12)** and [Cu(C-4oxy-acet)_2_(H_2_O)_4_] (**22**) in solution (DMSO) to assess to coordination geometry in solution and to simulate and explain the character and nature of the electronic transitions observed. The C_i_ symmetry and ^1^A_g_ ground state were suggested for the optimized geometry of **12** and **22** in solution (Figure 3). The calculations revealed that in solution the Zn/Cu-O_w_ bonds slightly decrease and the Zn/Cu-O_coum_ bonds increase by 0.02 Å. The pseudo-octahedral geometry of [Zn(C-4oxy-acet)_2_(H_2_O)_4_] complex is kept in solution and the uncoordinated oxygen forms hydrogen bonds with axial and equatorial aqueous hydrogens. In the crystal structure, where solid-state packing forces dominate, the carboxylic oxygen interacts mainly with the axial aqueous hydrogen. As expected for many Cu(d^9^) complexes, the [Cu(C-4oxy-acet)_2_(H_2_O)_4_] complex has axial elongated octahedral geometry (pseudo-octahedral) both in gas phase and in solution. In solution, the intramolecular hydrogen bond between the carboxylic oxygen and equatorial aqueous hydrogen atom stabilizes the coumarin and equatorial water ligands in the plane (Figure 3).

The electronic spectral data revealed that the observed absorption bands of the coumarin ligands (**1**–**10**) are in the range from 340 to 240 nm and they are assigned to one electron ππ* transition (Table 4). All these bands are kept in the corresponding Zn(II) and Cu(II) complexes without visible shifts (up to 10 nm). Only an increase of the molar absorptivity values (M^−1^ cm^−1^) was observed for the Zn(II) and Cu(II) complexes (Appendix A), possibly due to the polarisation in the ligands caused by the metal–ligand binding [38]. It is evident that the vertical excitation energies due to intraligand electron transitions are not affected by the metal ion in the complexes. A survey of the molecular orbitals involved in the transitions mentioned above showed an absence of the carboxylic carbon and oxygen orbitals, whose energies should be most influenced by ligand–metal interaction. In the absorption spectra of the Cu(II) complexes (**21**–**30**), one additional broad band was observed at 785–750 nm and it was assigned to d-d transition (Appendix A). In virtue of the molecular orbital and vertical excitation energy calculations of Cu(C-4oxy-acet)_2_(H_2_O)_4_ (**22)**, the unpaired electron occupies the d_x2-y2_ orbital with A_g_ symmetry and accordingly the open-shell orbital is d_x2-y2_ orbital (x and y axes are directed along the M-L bonds). This finding is in agreement with the previous suggestions for monodentate Cu(II) complexes with axial elongated octahedron [39]. Four spin-allowed d-d transitions with increasing energy are possible in the visible region: d_xy_ d_x2-y2_, d_yz_ d_x2-y2_, d_xz_ d_x2-y2_ or d_z2_ d_x2-y2_ for the Cu(II) complex. These transitions are forbidden by symmetry rules but obviously one of them is activated by vibro-electronic interaction and symmetry lowering.

### 2.6. Experimental and Theoretical ^1^H, ^13^C NMR Spectral Analysis of Zn(II) Complexes of 2-(2-oxo-2H-chromen-Substituted-yl)oxy Acetic Acid Ligands

Formation of Zn(II) complexes in solution was also confirmed by means of ^1^H and ^13^C NMR spectroscopy. The ^1^H NMR and ^13^C NMR spectral data of the selected Zn(II) complexes (**11**, **12**, **19**) compared to those of the corresponding ligands **1**, **2** and **9** are presented in Table 5 and Table 6, respectively. The stacked ^1^H NMR and ^13^C NMR spectra of [Zn(C-4oxy-acet)_2_(H_2_O)_4_] (**12)** and C-4oxy-acetH (**2)** are shown in Appendix A and Appendix A, respectively. To give reliable peak assignments and to confirm the suggested metal complex structure, ^1^H and ^13^C NMR chemical shift calculations were performed for selected [Zn(C-3oxy-acet)_2_(H_2_O)_4_] **(11),** [Zn(C-4oxy-acet)_2_(H_2_O)_4_] **(12)** and the corresponding ligands **1** and **2**. The obtained good agreement of the calculated and experimental NMR data confirmed the supposed model geometrical structures of the Zn(II) complexes in solution (Figure 3).

The ^1^H NMR peak at ~13 ppm in the NMR spectra of the oxy acetic coumarin ligands assigned to carboxylic hydrogen suggested an elongated O-H bond due to a strong solvation of this group. The model calculations showed that only simulation of the microsolvation around the COOH group (which is free by intramolecular hydrogen bond interaction) could reproduce such a downfield ^1^H NMR peak at ~13 ppm. The signal corresponding to the carboxylic acid exchangeable hydrogen was absent in the ^1^H NMR spectra of 2-(2-oxo-*2H*-chromen-substituted-yl)oxy acetic acid-based Zn(II) complexes, relative to the free ligands, suggesting metal binding through the carboxylate group (Appendix A). As expected, the ^1^H NMR spectra of the complexes showed upfield chemical shifts for most of the protons relative to the free acids with the largest upfield chemical shift observed for H^11^, which is next to the coordinating carboxylate group. A similar large shielding effect on H^11^ was observed for the Ag(I) complexes of the same series of ligands [7]. Hence it could be concluded that the upfield shift of H^11^ supposes coordination of the ligand through the carboxylate group, but this is not indicative of the coordination mode, monodentate or bidentate bridging. The aromatic protons are only slightly influenced upon complexation, which is consistent with their relative remoteness from the coordination centre.

The carbons C^12^ (of the carboxylate group) and C^11^ (near to the carboxylate group) exhibited the most significant downfield shifts (~2 ppm) in the Zn(II) complexes in agreement with the monodentate carboxylic oxygen coordination to the Zn(II) predicted by the molecular modelling in solution. It should be mentioned that a similar downfield shift was observed for C^11^, but not for C^12^ in the Ag(I) complexes with the same ligands, where bidentate bridging through carboxylic group was realized. Hence, we can conclude that a downfield chemical shift of C^12^ is characteristic for the monodentate carboxylate binding (Appendix A). The observed deshielding of C^11^ and C^12^ i.e., decrease of their electron density upon metal complexation, is due to the polarisation effect of Zn(II) cation or to ligand–Zn(II) electron transition (donor–acceptor L-M interaction). The ^13^C chemical shifts for the aromatic carbons were slightly shielded or deshielded.

### 2.7. Antimicrobial Studies

The antibacterial activity of the ligands used in the present study was studied in our previous investigation and no significant antibacterial or antifungal activity against the strains tested was found [7]. The Ag(I) complexes of these ligands, however, showed significant activity, particularly against the clinically relevant *C. albicans*, MRSA and *P. aeruginosa.* In addition, a number of coumarin- and quinolinone-derived Cu(II) complexes studied by our groups had previously shown excellent activity against *C. albicans* in particular. A recent series of studies showed that Zn(II) complexes have excellent activity against a number of bacterial strains and the Cu(II) complexes of coumarin derivatives have excellent antibacterial and antifungal activity [30,31]. That work, coupled with other studies on the antimicrobial activity of Cu(II) carboxylate complexes and analogous Zn(II) complexes in some cases, was the rationale for the preparation of the series of Cu(II) and Zn(II) complexes isolated here. Unexpectedly, all of the Zn(II) and Cu(II) complexes tested in this study showed no antimicrobial or antifungal activity against the strains tested (Appendix A). In our previous studies, the most active Cu(II) complexes were the ones with the best aqueous solubility. The Cu(II) complexes of coumarin derivatives isolated in the present study, although largely soluble in DMSO, also have sparing solubility in 50:50 water:ethanol mixture. The lack of activity of the Cu(II) and Zn(II) complexes studied here against all three species stands in stark contrast to the activity of their Ag(I) analogues. The role of Ag(I) complexes as antimicrobial agents has been postulated to be multifactorial, involving a number of possible mechanisms [40]. One mechanism is related to the controlled release of silver ions at the target site. The nature of the Ag(I) carrier, whether nanoparticle or ligand, can then be tuned to target the active site or pathogen. If release of the metal ion is the basis of this series of complexes activity, then the complexes isolated here may not release their metal ion due to the difference in coordination mode of the Ag(I) complexes to the Cu(II) and Zn(II) complexes studied. In the Ag(I) complexes, the oxyacetate ligands bind via a chelating carboxylate mode but in the complexes isolated here, only unidentate bonding is noted [40]. The difference in bonding mode may well have an effect on the release of the metal ion. A more recent review on the antimicrobial role of Cu(II) complexes postulates three possible mechanisms of action for antimicrobial activity including: (1) Cu(II) complexes undergoing Fenton-like reactions to generative reactive oxygen species; (2) eliciting an inflammatory response which targets pathogens; or (3) the release of free Cu(II) or Cu(I) from the complex at the target site again resulting in Fenton-like chemistry in the presence of intracellular co-factors or else in the form of copper nanoparticles targeting cell membranes of pathogenic species [41]. Many previous studies in particular have shown that copper ions are toxic to microbial species and a recent review indicates that release of Cu(II) and reduction in situ to Cu(I) can damage the membrane and infiltrate the cell, inducing an oxidative stress response involving endogenous ROS [41,42].

The antimicrobial role of zinc complexes has focused much more on Zn-nanoparticles, as reviewed by Almoudi et al., but studies on zinc complexes indicate that the role of Zn(II) lies in interference in metabolic pathways within the microbial cell [43]. We have also previously speculated that the antimicrobial activity of Cu(II) complexes may be related to their ability to act as biomimetics and all previous copper coumarin complexes investigated by us, which showed biomimetic activity, did not have octahedral structures [30,44,45]. The active Zn(II) carboxylate complexes isolated by other groups have a variety of geometries and coordination environments although many, including those previously isolated by our group, had octahedral geometry [46]. The lack of activity of the complexes isolated here may be a result of the coordination modes for the central metal ions, which mean that the complexes lack the ability to act as biomimetics or that they are simply not getting into the microbial cells. Further mechanistic studies would be needed to elucidate this and a comparison between these and active Cu(II) and Zn(II) complexes in terms of biomimetic ability, localisation in cells etc. may help identify a new therapeutic target for metal-based complexes.

## 3. Materials and Methods

### 3.1. Experimental Methods

Chemicals and solvents were purchased from Sigma-Aldrich Co. (Dorset, UK) and used without further purification. Melting points were recorded on a Stuart Scientific SMP-1 apparatus (up to 300 °C). Thin layer chromatographic (TLC) was carried out on Silica gel 60F_254_ aluminium-backed plates using a pre-saturated elution tank. The compounds were detected under ultraviolet light and stained using iodine crystals in a closed chamber. Infrared spectra of solids (in KBr disc) were recorded on a Nicolet Avatar 320 FT-IR spectrophotometer operating at a resolution of 2 cm^−1^ in the region 4000–400 cm^−1^. Bruker Biospin 300 and 500 MHz FT-NMR spectrometers were used to record ^1^H NMR spectra (-5 to 15 ppm with a resolution of 0.18 Hz) and ^13^C NMR spectra (-33 to 233 ppm with a resolution of 2.40 Hz) as solutions in *d_6_*-DMSO, using tetramethylsilane (TMS) as the internal reference standard. The signal assignments were made using standard techniques including DEPT, COSY and CH-Shift and where necessary HSQC, HMBC and NOESY experiments were also run to facilitate peak assignments. Microanalytical data were provided by the Microanalytical Laboratory, Department of Chemistry, University College Dublin, Belfield, Dublin 4, Ireland, using a Perkin-Elmer CHN analyser. The metal (zinc and copper) content was determined using atomic absorption spectroscopy on a Perkin Elmer 460AAS instrument. The zinc and copper standards and samples were analysed at 213.90 and 324.70 nm, respectively. All samples were analysed in triplicate. UV-Vis spectra of the complexes and their corresponding ligands were recorded on a MSC UV-T80 spectrophotometer in dimethyl sulfoxide (DMSO) solvent (260–800 nm). The molar conductivity of the complexes and their corresponding ligands was measured on an EUTOH INSTRUMENTS Cyberscan-500 digital conductometer, using 4 mM solution of compounds in DMSO at 25 °C. Solid-state magnetic susceptibility measurements for Cu(II) complexes were carried out at room temperature using a Johnson Matthey Magnetic Susceptibility Balance with [HgCo(SCN)_4_] as a reference standard, in the Chemistry Department, Maynooth University, Co. Kildare, Ireland.

A single crystal was coated in Paratone-N heavy oil then mounted on a Hampton Research Cryoloop and placed in a cold stream of nitrogen gas (100 K) on a Bruker X8 Apex2 CCD diffractometer running the Apex2 software. Data collection consisted of, on average, 3000 frames from a combination of *phi* and *omega* scans. After unit cell determination, intensity data were integrated using SAINT then scaled with SADABS (TWINABS if the crystal was not single) [47]. OLEX2 [34] with the SHELX suites of programs were used for structure solution and refinement. Pictures were prepared using the graphics program XP [48,49]. The quality of the dataset was checked using the IUCr checkcif facility. Crystallographic data have been deposited at the Cambridge Crystallographic Data Centre; 2253752 (**12**) and 2253753 (**19**).

### 3.2. Computational Details

Molecular structures, vibrational (IR), NMR and absorption spectra calculations of the model Zn(II) complexes Zn(C-3oxy-acet)_2_(H_2_O)_4_ (**11**), Zn(C-4oxy-acet)_2_(H_2_O)_4_ (**12**) and Cu(C-4oxy-acet)_2_(H_2_O)_4_ (**22**) were carried out with DFT/TDDFT method using Gaussian16 program code [50] and the geometrical and spectroscopic data were visualized by the ChemCraft program [51]. The X-ray structure for Zn(C-4oxy-acet)_2_(H_2_O)_4_ (**12**) was used as an input geometry for calculations of this complex at molecular level. By analogy, a monodentate carboxylate binding of the coumarin ligand was simulated in the model complexes **11** and **12**, with general formula M(L)_2_(H_2_O)_4_ M = Zn(II), Cu(II). For comparison, the calculations were performed for the corresponding isolated neutral ligands C-3oxy-acetH (**1**) and C-4oxy-acet (**2**). Split valence plus polarisation basis set 6–31G(d) was applied for the main group elements. To give a better description of the wave functions in the intermolecular region and hence to improve the estimation of the Zn(II)/Cu(II)–coumarin ligand interactions, diffuse functions were added to the standard basis set (one s and one p set) for all O atoms as well as for the two acetoxy carbon, C11 and C12, atoms (see a scheme under Appendix A). For Zn and Cu, 6-31++G(d) basis set was employed [52]. The combined basis set, denoted as CB1**,** was used in combination with the Lee, Yang, and Parr correlation functional (LYP) [53] and Becke’s three parameters exchange functional (B3) [54]. The adequacy of the B3LYP method for prediction of geometry parameters and vibrational spectra of coumarin derivatives and their metal complexes was proven in our previous investigations [55,56].

The ^1^H and ^13^C NMR chemical shift calculations of the model compounds **1**, **2, 11** and **12** were carried out in DMSO at B3LYP/6-31+G(d,p) and B3LYP/6-31++G(2d,p) levels of theory. The UV-Vis absorption spectra of the ligands and their Cu(II) and Zn(II) complexes were simulated in the region 200–400 nm by means of vertical excitation energy calculations in DMSO using TDDFT/B3LYP/CB1 method. NMR and UV–Vis spectra calculations in DMSO were carried out with the optimized geometries in the same solvent which was simulated by the Polarisable Continuum Model (PCM) [57,58,59].

For reliable interpretation of the experimental solid-state infrared spectra, harmonic vibrational wavenumbers of the simulated crystallographic structures **12** and **19** were computed. The periodic DFT calculations were performed using the Vienna Ab initio simulation package (VASP) [60]. Electron exchange correlation interactions were treated using the generalized gradient approximation as parametrized by Perdew, Burke and Ernzerhof [61]. The electron-ion interactions were described using the projector-augmented-wave (PAW) method [62]. The vibrational frequencies were calculated using density functional perturbation theory (DFPT) (linear response theory) [63,64,65].

### 3.3. Antimicrobial Studies

The ligands (**1**–**10**), Zn(II) and Cu(II) complexes (**11**–**30**) and the commercially available drug vancomycin were screened for their antimicrobial activity. The activity of the ligands has been reported previously [7]. *S. aureus* (NCIMB 12702) was obtained from the National Collection of Industrial, Marine and Food. *MRSA* strain ATCC4300 and *P. aeruginosa* strain ATCC 27853 were obtained from the American Type Culture Collection. *P. aeruginosa* were cultured in Luria Bertani (LB) broth (Sigma) while *MRSA* was cultured in nutrient broth. Bacteria were stored in broth with 20% glycerol at -80 ^o^C and passaged twice on agar plate (LB agar, TSA or nutrient agar respectively) before being used in an experiment. MIC_50_ were determined by broth microdilution. Stock solutions of each compound to be tested were prepared in DMSO and serially diluted in triplicate using broth (100 µL/well) at a concentration range of 0.195–200 µM on microtitre plates, after which the final concentration of DMSO in the cell suspension was not greater than 1%. Overnight cultures (10 mL) of bacterial strains were inoculated in broth (100 mL) and incubated at 37 ^o^C with agitation, optical density at 600 nm (OD_600_) was measured and strains were then diluted to 1 x 10^6^ CFU/mL and seeded at 100 µL/well in microtiter well plates. Bacteria were also treated with a solvent control (0.5% DMSO) in control wells. Following 24 h incubation, the absorbance was read in a microtiter plate reader. Each compound was screened at each concentration at least in triplicate and three independent experiments were carried out.

The isolated ligands of these complexes were tested previously under the same conditions and with the same strains as their metal complexes [7]. All complexes were tested for their anti-*Candida* activity using a broth microdilution susceptibility protocol established by the National Committee for Clinical Laboratory Standards (NCCLS), document M27-A2, with slight modifications. The M27-A2 method was altered by substituting antibiotic medium 3 (Oxoid Ltd.) for RPMI 1640 medium [66]. All of the test compounds were prepared in 2% DMSO solution as the compounds were soluble only in DMSO. The maximum concentration of the DMSO, after serial dilution, in the test wells was 0.5%. The inhibitory effect of DMSO on the growth of *C. albicans* was also examined and it was shown that 0.5% DMSO did not affect the growth of this particular strain (ATCC 10231) of *C. albicans*. The ligands, complexes and a commercially available antifungal drug Amphotericin B were tested against *C. albicans* (ATCC 10231). The screening was carried out according to the broth microdilution reference method [66].

## 4. Conclusions

The preparation of novel Zn(II) (**11**–**20**) and Cu(II) (**21**–**30**) complexes of coumarin-derived oxyacetate ligands (**1**–**10**) was undertaken to improve the solubility profile of coumarin-derived Cu(II) and Zn(II) complexes towards increasing the antimicrobial activity of this type of complex. The complexes were synthesized and characterized in detail by both experimental and computational methods. The new Zn(II) and Cu(II) complexes correspond to the general formula [M(L)_2_(H_2_O)_4_]_x_nH_2_O and form pseudo-octahedral geometries. The solid-state structures of two Zn(II) complexes [Zn(C-4oxy-acet)_2_(H_2_O)_4_] (**12**) and [Zn(4CF_3_-C-7oxy-acet)_2_(H_2_O)_4_] (**19**) were determined by X-ray crystallographic measurements, which revealed a monodentate carboxylate binding of the two coumarin ligands to the Zn(II) in the monomer complexes. The experimental and calculated solid-state IR, solution UV-Vis and NMR spectra predicted similar ligand coordination behaviour in all the studied Zn(II) and Cu(II) complexes. Unfortunately, none of the newly synthesized Zn(II) and Cu(II) complexes of coumarin oxyacetate ligands displayed the expected antimicrobial activity against the clinically relevant MRSA, *Pseudomonas Aeruginosa* and *Candida albicans*. A comparative analysis of the coordination behaviour of the ligands and the resulted coordination polyhedron in the solid state and solution was carried out to explain the unexpected lack of biological activity of the studied Zn(II) and Cu(II) complexes. DFT structure modelling in solution predicted molecular structures for the Zn(II) and Cu(II) complexes which are in very good agreement with their solid-state coordination geometries, indicating that the solid-state structure of these complexes is preserved in solution. This structural behaviour is different from that of other Zn(II), Cu(II) and Ag(I) complexes of coumarin ligands, studied by us previously, which have shown quite different ligand coordination to the metal centre in solution and in the solid state (in particular extensive contacts to Ag(I) ion), but have revealed excellent antimicrobial activity. In the complexes isolated here, octahedral coordination geometry is maintained in solution and that coordination sphere may hinder the biomimetic activity of the complexes or hinder their incorporation into the target cells. A SARS study could not be completed but a comparison of biomimetic activity of these complexes relative to previous coumarin-based Cu(II) and Zn(II) antimicrobial complexes isolated by our group may help elucidate the mechanism of action of this type of complex and its therapeutic target.

## Data Availability

Data is contained within the article or Appendix A.

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
