# Peer review of "Structural and Spectroscopic Study of New Copper(II) and Zinc(II) Complexes of Coumarin Oxyacetate Ligands and Determination of Their Antimicrobial Activity"

_molecules, 2023, doi:10.3390/molecules28114560_

Round 1

Reviewer 1 Report

The manuscript summarizes results collected via fairly routine experiments. The ligands are not new compounds, and also the synthesis of the complexes does not seem that it would have been a big challenge for researchers skilled in synthetic work. As a consequence, the novelty of the results collected in this part of the study is quite low.

The considerations, which were made during the selection of substituents on the coumarine-derivatives should be mentioned in the introductions

The characterization of the synthesized complexes (both the chemical and antimicrobial activity) is correct. Adequate methods were used and the interpretation of the results seems correct, as well.  

The biological experiments have provided interesting result. Namely, there is a big difference (contrast) between the antimicrobial activities of these complexes (inactives) compared to the similar silver(I) complexes and other coumarine-derivatives with copper(II) and zinc(II) (highly actives).   

Reviewer 2 Report

This submission by M Mujahid and co-workers describes the structural and spectroscopic study of new copper(II) and zinc(II) complexes of coumarin oxyacetate ligands and determination of their antimicrobial activity. In general, the research is designed appropriately, and the methods are adequately described. Some results are clearly presented and the conclusion supported by the experimental results. I recommend the publication of the paper in the Molecules magazine after major revisions given as follows:

 1. The picture resolutions of Figs. S6 and S7 should be better.

2. Copper(II) complexes are not fully characterized, especially not supported by X-ray crystallography. Please provide the 1H NMR and 13C NMR of some copper complexes as the supplementary materials.

3. The purity of the product was tested by appropriate methods. It is well known that the purity of synthetic compound can strongly affect the experimental results.

4. The stability of the Zn(II) and Cu(II) complexes should be studied also in a buffered solution with a low percentage of DMSO if solubility is a problem.

No

Reviewer 3 Report

The manuscript gives a detailed characterization of Zn(II)- and Cu(II)-complexes with coumarin oxyacetate, which is and solid data. However, it would be nice to have a discussion of the differences between the previously reported Cu(II)- and Zn(II)-complexes, that had antimicrobial activity. 

Author Response

Please see that attachment

Reviewer 4 Report

1.      Abstract. The introductory part was complete ignored, the manuscript has no continuous lines, so it is difficult to review it,  and by x-ray crystallography for two of the zinc complexes whereas Cu complexes were also reported but nothing was explained about these complexes, why only Zn complexes

2.      Abstract: which has not been the case in our previous studies of analogous coumarin silver(I) complexes, this sentence has no connection with Zn and Cu complexes, it can be explained in “Result and Discussion”, it is inappropriate

3.      Abstract: Previous studies had indicated excellent antimicrobial activity for silver(I) analogues,…again this sentence is irrelevant and explain here what the authors find in the present study, and also conclude the findings of present research

4.      Introduction: The importance of coumarin shall be explained before the authors’ previous studies

5.      Introduction: The name of bacteria and fungus shall be abbreviated first then use it throughout the manuscript and same is the case with other words used subsequently as abbreviated

6.      Introduction: Explain the previous studies which were performed with Zn and Cu complexes for the antimicrobial activities based on coumarin and related analogs rather to explain the complexes of Zn and Cu with Schiff bases and any other, this is not a good rationale for the design of present study.

7.      If the Zn and Cu complexes with coumarin based ligands are not reported for the strains mentioned in this study, then it will be appropriate

8.      The zinc(II), and copper (II), it shall be Zn (II) and Cu (II), correct it throughout the manuscript

9.      What is the problem of the previous studies that need to be solved? What is the meaning and innovation of this work? The advance of this work compared with other works should be given in detail to rationalize the present work. The rationale of present research is poor, improve it.

10.    The limitations and strength of present study was not explained

11.    The 90 % of the manuscript was focused on spectroscopic studies, it seems it just explain the spectroscopic behavior of new design Zn and Cu complexes, a small portion was explained for antimicrobial activities

12.    No quantitative data of antimicrobial activities was explained

13.    Now two options, 1. Totally explain the spectroscopic properties of newly design complexes and change the title and rationalize the study accordingly, option 2. The existing title remain intact the added the comprehensive data about the antimicrobial activities, what was the outcome of present studies for the antimicrobial activities

14.    Then explain the limitations of present research and innovations for the further studies to find new horizons

15.    No findings were explained in the conclusion about the screening studies for bacterial and fungus strains, conclusion is very poor.

16. 50 % references are older than last 5 years 

Moderate editing of English language

Author Response

Please see file attached

Round 2

Reviewer 2 Report

The revised version is now suitable for publication.

No

Reviewer 4 Report

The content of paper was well organized, all the suggested points are incorporated and easy for the reader to follow the subject discussed, thus support for its acceptance.

 Minor editing of English language required